# Alterations in Primary Carbon Metabolism in Cucumber Infected with *Pseudomonas syringae* pv *lachrymans*: Local and Systemic Responses

**DOI:** 10.3390/ijms232012418

**Published:** 2022-10-17

**Authors:** Tomasz Kopczewski, Elżbieta Kuźniak, Iwona Ciereszko, Andrzej Kornaś

**Affiliations:** 1Department of Plant Physiology and Biochemistry, Faculty of Biology and Environmental Protection, University of Lodz, 90-237 Lodz, Poland; 2Department of Plant Biology and Ecology, Faculty of Biology, University of Bialystok, 15-245 Bialystok, Poland; 3Institute of Biology, Pedagogical University of Krakow, 30-084 Kraków, Poland

**Keywords:** angular leaf spot disease, malic acid, plant–pathogen interaction, pyridine nucleotides, raffinose, soluble sugars, TCA cycle

## Abstract

The reconfiguration of the primary metabolism is essential in plant–pathogen interactions. We compared the local metabolic responses of cucumber leaves inoculated with *Pseudomonas syringae* pv *lachrymans* (*Psl*) with those in non-inoculated systemic leaves, by examining the changes in the nicotinamide adenine dinucleotides pools, the concentration of soluble carbohydrates and activities/gene expression of carbohydrate metabolism-related enzymes, the expression of photosynthesis-related genes, and the tricarboxylic acid cycle-linked metabolite contents and enzyme activities. In the infected leaves, *Psl* induced a metabolic signature with an altered [NAD(P)H]/[NAD(P)^+^] ratio; decreased glucose and sucrose contents, along with a changed invertase gene expression; and increased glucose turnover and accumulation of raffinose, trehalose, and *myo*-inositol. The accumulation of oxaloacetic and malic acids, enhanced activities, and gene expression of fumarase and l-malate dehydrogenase, as well as the increased respiration rate in the infected leaves, indicated that *Psl* induced the tricarboxylic acid cycle. The changes in gene expression of ribulose-l,5-bis-phosphate carboxylase/oxygenase large unit, phospho*enol*pyruvate carboxylase and chloroplast glyceraldehyde-3-phosphate dehydrogenase were compatible with a net photosynthesis decline described earlier. *Psl* triggered metabolic changes common to the infected and non-infected leaves, the dynamics of which differed quantitatively (e.g., malic acid content and metabolism, glucose-6-phosphate accumulation, and glucose-6-phosphate dehydrogenase activity) and those specifically related to the local or systemic response (e.g., changes in the sugar content and turnover). Therefore, metabolic changes in the systemic leaves may be part of the global effects of local infection on the whole-plant metabolism and also represent a specific acclimation response contributing to balancing growth and defense.

## 1. Introduction

The plant immune system comprises both constitutive and induced defense responses. The constitutive defense is based on preformed structural barriers, such as a waxy epidermal cuticle, rigid cell walls, and antimicrobial secondary metabolites [1,2,3]. The induced defense responses, activated by recognition of pathogen-associated molecular patterns (PAMPs) through plant cell surface-anchored pattern recognition receptors (PRRs), form the pattern-triggered immunity (PTI). Those activated by the recognition of effector proteins, translocated by the pathogen to the plant cells through intracellular nucleotide-binding domain leucine-rich repeat-containing receptors (NLRs) constitute the effector-triggered immunity (ETI) [4,5,6]. Pathogen invasion results in disease development when the immune system is not activated or is triggered too late to prevent the progression of microbial colonization. Disease susceptibility, however, is more than a failure of host immunity. It requires active host cooperation, since pathogens recruit plant reactions to transform the host into an ecological niche supporting successful infection [7].

Pathogen infection causes reconfiguration of the primary plant metabolism, contributing both to defense responses and the establishment and progression of the disease. According to the growth–defense concept, defense depends on the primary metabolites and energy is shifted from growth, reproduction, and storage processes [8]. After a pathogen attack, the extensive reconfiguration of the plant primary metabolism serves multiple defense purposes. First, it directly supports carbon and energy requirements for defense responses. It also balances competitive growth and defense processes, to optimize plant fitness under biotic stress, and provides regulatory signals conferring plant resistance to pathogens [9]. In the susceptible interactions, where the disease progresses, metabolic changes may result from the pathogen strategy to manipulate the plant metabolism to obtain nutrients from host cells [10].

The metabolic changes in various plant interactions with biotrophic, hemibiotrophic, and necrotrophic pathogens fit into a model whereby the pathogen induces a rapid local decline in photosynthesis and an increase in respiration, photorespiration, cell wall-bound and vacuolar invertase activities, and elevated expression of SWEET (sugars will eventually be exported) sugar transporters, indicating the induction of sink status in the infected tissues [7,11,12]. These metabolic changes were detected in susceptible and resistant interactions, but they occurred earlier and/or to a greater extent in the resistant ones [12,13,14]. Although infection-induced modifications in photosynthesis and carbon metabolism are complex and depend on specific plant–pathogen interactions, appropriately maintaining photosynthesis during infection seems crucial for defense and plant fitness. These metabolic adjustments are usually related to the pathogen-induced alterations in the source-sink relationships in the infected plant. In interactions with several pathogens, the inhibition of photosynthesis is often confined to the infection sites, whereas in plant tissues surrounding the infected areas and in uninfected distal leaves, it is upregulated or unchanged [1,12,15].

Biotrophic pathogens whose growth and reproduction depend on plant-derived carbon sources, including carbohydrates, have been suggested to primarily obtain nutrients from photosynthetically-active, uninfected neighboring cells [16]. For the host plant, however, the shift from assimilatory to non-assimilatory and carbohydrate-consuming metabolism may be a prerequisite for the induction of plant defense responses. In tobacco, a complete decline in photosynthesis at the site of *Phytophthora nicotianae* infection preceded defense responses related to the hypersensitive response (HR) and callose deposition. The collapse in photosynthesis was accompanied by an increase in respiration and oxidative pentose phosphate pathway (OPPP) [12]. Downregulation of photosynthesis in infected leaves may also lead to nitrogen release from ribulose-1,5-bisphosphate carboxylase-oxygenase (RuBisCO), which can be used to produce nitrogen-containing compounds playing a defensive role in plants [17].

In several plant–pathogen interactions, an indication of the sink status of infected leaves is the increased activity of apoplast invertase [7,14]. Invertases cleaving sucrose into fructose and glucose regulate carbohydrate partitioning; the sink strength and the source/sink relations that are essential in disease processes [18]. Carbohydrates are the primary substrates providing energy for the defense reactions and carbon skeletons for the synthesis of defense compounds. Moreover, saccharides, such as sucrose, fructose, glucose, and trehalose, act as signals inducing defense genes, and they probably function as priming molecules inducing PTI and ETI in plants [19,20,21]. Hexose signals trigger the induction of *PR1* (pathogenesis related) and *PR3* gene expression [22]. A recent study, however, revealed that the role of invertases in plant–pathogen interactions depends on the lifestyle of the pathogens, and invertases may act as resistance and susceptibility factors in response to hemibiotrophic and biotrophic pathogens, respectively [23].

An expression analysis of genes encoding the transcription factors and metabolic enzymes in the model plant *Arabidopsis thaliana* under pathogen- and elicitor-induced biotic stress revealed that the tricarboxylic acid cycle (TCA), glycolysis, OPPP, mitochondrial electron transport, and biosynthesis of ATP were upregulated and constituted key processes delivering energy for defense [24,25]. In tomato, hexanoic acid-induced resistance to *P*. *syringae* was associated with a significant reduction in the TCA pathway, suggesting that plant defense drains the TCA intermediates [26]. In soybean infected with *P*. *sojae*, a metabolomic analysis identified many sugars, amino acid derivatives, and organic acids differentially accumulated in the resistant plants compared to the susceptible ones, suggesting that they may participate in the defense response [27].

Host metabolic responses to pathogen infection are mostly linked to processes integrating carbon and nitrogen metabolism with energy production. Pyridine nucleotides, NAD(H) and its phosphorylated form NADP(H), apart from their role in the redox regulation of plant metabolism and signaling pathways [28,29], modulate the integration of carbon and nitrogen metabolism [30]. Moreover, the intracellular balance between NAD(H) and NADP(H) interferes with plant defense responses. For example, the NAD^+^ cleavage enzymes negatively influence plant resistance [29], and extracellular NAD(H) and NADP(H) induce PR gene expression and resistance to *P*. *syringae* pv *maculicola* in *Arabidopsis* [31].

These pathogen-induced metabolic changes may also serve disease propagation by transforming the hostile intact plant into an active constituent of the pathosystem [32]. Gram-negative bacterial pathogens secrete effector proteins into host cells to hijack plant metabolism for the biosynthesis of nutrients supporting microbial growth and tissue colonization [10]. In *Arabidopsis* infected with virulent *P*. *syringae*, metabolic reprogramming during disease establishment, superimposed on defense suppression, led to changes in the levels of sugars and purine amino acids, supporting the growth of the bacterial population in the apoplast [33]. *P*. *syringae* pv *actinidiae*, a causative agent of bacterial canker disease in kiwifruit, enhanced infection by inducing extensive C and N reprogramming via inhibiting photosynthesis, C fixation, and the TCA cycle [34].

While pathogen-induced local primary metabolism changes in infected tissues have been widely documented [7,24,25], the systemic response in uninfected ones has received comparatively less attention. At the organism level, infected plants coordinate immune responses, and the leaf position-dependent variation in stress reactions was suggested to be an active mechanism important for balancing defense and plant growth [35].

In our previous study with cucumber—the *P*. *syringae* pv *lachrymans* (*Psl*) pathosystem—we found a pathogen-induced decline in the photochemical activity of PSII, stomatal conductance, and photosynthetic gas exchange at the advanced stage of pathogenesis. These changes occurred in the infected leaves, along with full disease symptom development, and not in the pathogen-free systemic leaves and were also related to redox regulations [1].

This study aimed to examine the metabolic changes in (1) the nicotinamide adenine dinucleotide pools; (2) the contents of soluble carbohydrates and activities/gene expression of carbohydrate metabolism-related enzymes; (3) the expression of photosynthesis-related genes; and (4) the respiration rate and the TCA-linked metabolite contents and enzyme activities. These factors are involved in integrating carbon metabolism with energy production and were predicted to have relevance for plant responses to pathogens [26,34,36,37,38]. We compared the metabolic changes of an infected, fully expanded third cucumber leaf with that of an uninfected, expanding fifth leaf, concerning the timing, quantity, and quality of the responses.

## 2. Results

In our experimental system, the *Psl* infection-induced angular leaf spot disease symptoms, in the form of angular necrotic lesions with extensive chlorosis (Figure A1), were restricted to the inoculated third leaf (L3 + *Psl*) and never spread to the other leaves and the stem [1].

### 2.1. Respiration Rate

The infection-induced changes in respiration rate patterns differed in the L3 + *Psl* and L5 + *Psl*. The respiration rate (respiratory consumption of O_2_ by leaves) increased considerably in L3 + *Psl* at 2 and 7 day (days after inoculation), up to 156% and 150% of the respective controls, whereas it did not change in L5 + *Psl* (Table 1).

### 2.2. Nicotinamide Adenine Dinucleotides [(NAD(P)H, NAD(P)^+^] Contents

Infection increased the [NADH]/[NAD^+^] redox ratio in L3 + *Psl* compared to the control at 5 and 7 days. This was associated with higher NADH content (40–50%) and a decrease in NAD^+^ content (about 45%) in L3 + *Psl*. Moreover, about 70% [NADH]/[NAD^+^] higher redox ratio in L5 + *Psl*, compared to L5 only at 7 day was observed (Figure 1A, Table A1). In contrast to changes in the NADH pool, a decrease in NADPH content in L3 + *Psl*, compared to the control at 1–7 day, was observed. Moreover, in L3 + *Psl*, an increase of NADP^+^ (about 60% compared to the control) was detected at 5 and 7 day. A similar trend of post-infectious changes was observed in L5 + *Psl* in the late stages of infection, when a decrease in the [NADPH]/[NADP^+^] ratio compared to L5 was detected (Figure 1B, Table A1).

### 2.3. Saccharides and Myo-inositol Contents

*Psl* infection diminished the content of d-glucose in L3 + *Psl* compared to the control at 2–7 day, while simultaneously increasing that of d-fructose (Figure 2A,B). The most significant difference (about 45% compared to the control) was found at 7 day. The post-infectious increase in d-fructose content in L3 + *Psl*, compared to L3, was coupled with a decrease in sucrose content at 5–7 day. In the L5 + *Psl*, the saccharides contents did not change, except for an increase in sucrose by 28%, compared to L5 at 5 day (Figure 2C).

The contents of other non-reducing saccharides: trehalose and raffinose, also changed after *Psl* infection. The trehalose content increased twice, both in L3 + *Psl* and in L5 + *Psl*, compared to the appropriate control at 7 day. Raffinose was elevated at 2 and 7 day; at 2 day, it was 28-fold higher in L5 + *Psl* than in L5. The content of *myo*-inositol was increased in L3 + *Psl* at 2 and 7 day compared to the control (Table 2).

### 2.4. Acid (AcInv) and Alcaline (AlInv) Invertases Genes Relative Expressions

As a result of *Psl* infection, the *AcInv* expression level in L3 + *Psl* was about sixfold lower compared to the control. Similar changes were found in L5 + *Psl* only in the early stage of pathogenesis; four- and seven-fold reductions of *AcInv* relative expression in L5 + *Psl*, compared to L5 was demonstrated at 1 and 2 day, respectively (Figure 3A). The changes in *AlInv* relative expression in L3 + *Psl* differed from those found for *AcInv*. We found 10-, 8-, and 6-fold higher levels of *AlInv* expression in L3 + *Psl*, compared to L3 at 1, 2, and 7 day, respectively (Figure 3B).

### 2.5. Glucose-6-phosphate Content and Glucose-6-phosphate Dehydrogenase (G6PDH) Activity. Hexokinase (HK) and G6PDH Genes Relative Expressions

The post-infectious decrease in d-glucose content in L3 + *Psl* at 2, 5, and 7 day (Figure 2A) was paralleled by an increase in glucose-6-phosphate content, compared to the control. In the L5 + *Psl* leaves, *Psl* infection caused a significant increase (by 45%) in glucose-6-phosphate content only at 7 day (Figure 4A). We also found that the G6PDH activity was about 80% higher in L3 + *Psl* compared to the control at 2–7 day. In L5 + *Psl*, G6PDH activity was about 30% higher than in L5 only at 7 day (Figure 4B). Post-infectious changes in *HK* relative expression occurred only in older leaves; a fivefold higher *HK* expression in L3 + *Psl* than in L3 was found (Figure 4C). Similarly, an increase in the *G6PDH* relative expression was shown in L3 + *Psl*, compared to L3 (about 9-, 8-, and 5-fold at 1, 2, and 7 day, respectively). However, the *G6PDH* expression level in L5 + *Psl* was about sixfold lower than in L5 at 1–7 day (Figure 4D).

### 2.6. l-Malic and Oxaloacetic Acids Contents, l-Malate Dehydrogenase (l-MDH) and Fumarase (FUM) Activities

After *Psl* infection, we observed an increase of l-malic acid content in L3 + *Psl* (2–7 day) and L5 + *Psl* (5 and 7 day), compared to the appropriate controls (Figure 5A). The profile of oxaloacetic acid content in L3 + *Psl* also changed after infection. It decreased at 2 day, but then there was an increase at 5 and 7 day, by 28% and 26%, respectively, compared to the control (Figure 5B). The post-infectious accumulation of l-malic acid in L3 + *Psl* was accompanied by an increase in l-MDH activity, compared to L3 (from 29% at 2 day to 41% at 7 day). The l-MDH activity in L5 + *Psl* did not differ significantly from that in L5, except for at 7 day, when it was enhanced (Figure 5C). In the late stage of pathogenesis (5 and 7 day), the FUM activity in L3 + *Psl* was higher by 43% compared to L3. Moreover, a higher FUM activity was also found in L5 + *Psl* compared to L5 at 7 day (Figure 5D).

### 2.7. l-Malate dehydrogenase (l-MDH), Fumarase (FUM), and Isocitrate Dehydrogenase (ICDH) Gene Relative Expressions

The post-infectious increase of l-MDH activity in L3 + *Psl* at 2–7 day (Figure 5C) was accompanied by an increase in the *l*-*MDH* relative expression in L3 + *Psl*, which was approximately ninefold higher than in L3. Moreover, we observed a threefold lower *l*-*MDH* relative expression in L5 + *Psl*, compared to L5 at 1–7 day (Figure 6A). A progressive increase of *FUM* expression in L3 + *Psl* compared to L3 was found, e.g., it was 10-fold higher in L3 + *Psl*, compared to the control at 7 day. However, the *FUM* relative expression in L5 + *Psl* decreased in comparison to L5. It was threefold lower than the control at 2 day (Figure 6B). The biotic stress induced by *Psl* infection did not change the relative expression of *ICDH*. Due to distinct differences between L3 and L5, the *ICDH* relative expression in L3 + *Psl* and L5 + *Psl* varied significantly in favor of the systemic leaves (Figure 6C). The opposite relationship was observed for the *l*-*MDH* and *FUM* relative expression, which was much lower in L5 + *Psl* than in L3 + *Psl* (Figure 6A,B).

### 2.8. Ribulose-1,5-bisphosphate Carboxylase/Oxygenase Large Unit (RuBisCO_LU_), Phosphoenolpyruvate Carboxylase (PEPC), Glyceraldehyde-3-phosphate Dehydrogenase (3PGAD), and Chloroplastic Thioredoxin M3 (TrxM3) Gene Relative Expressions

In the L3 + *Psl* leaves, the *RuBisCO_LU_* relative expression decreased four- and two-fold in the early phase of pathogenesis (1 and 2 day, respectively). The opposite trend was found in L5 + *Psl*, where the expression level of *RuBisCO_LU_* was three-times higher than in L5 (Figure 7A). Unlike *RuBisCO_LU_*, the *PEPC* relative expression in L3 + *Psl* was approximately fivefold higher than in L3 throughout the experiment. A similar relationship after infection was found in L5 + *Psl*. At 1 and 7 day, the *PEPC* expression levels were two- and three-fold higher in L5 + *Psl* than in L5 (Figure 7B). Post-infectious changes in the expression level of *3PGAD* in L3 + *Psl* were shown only at 7 day (threefold increase relative to L3) and in L5 + Psl at 2 day (twofold increase relative to L5) (Figure 7C). The relative expression of the chloroplast TrxM3 was 5- and 10-fold higher in L3 + Psl, compared to the appropriate control at 2 and 7 day, respectively. There were no significant differences between L5 + Psl and L5 regarding the TrxM3 expression level (Figure 7D).

## 3. Discussion

Plant primary metabolism is an essential element in plant–pathogen interactions. Metabolic changes are ubiquitous for many pathosystems; however, those induced locally within the infected area may differ from the response in the distant, non-infected tissues. These differential local and systemic responses allow plants to allocate resources to defense, while maintaining growth and reproduction [36]. In recent years, understanding the pathogen-induced changes in the primary metabolism at the organism level has attracted considerable attention concerning plant breeding, to improve the pathogen resistance and productivity of plants grown under biotic stress [39].

### 3.1. Local and Systemic Changes in the Nicotinamide Adenine Dinucleotides Pool

NAD(H) and NADP(H) are global metabolic regulators in plants, and maintaining the NAD(H) and NADP(H) balance is essential for the functioning of the cell [40]. The primary role of NAD(H) is related to ATP synthesis in mitochondria, whereas NADP(H) provides the reducing power for biosynthesis, signaling, and redox regulation. These distinctive functions of NAD(H) and NADP(H) are reflected by the ratio of reduced and oxidized forms, with [NADPH]/[NADP^+^] > 1.0 and [NADH]/[NAD^+^] < 1.0 [41]. NAD is highly oxidized in plants, whereas NADP mainly exists in the reduced form [42]. In our study, *Psl* infection shifted the [NADH]/[NAD^+^] ratio towards a more reducing state and the [NADPH]/[NADP^+^] ratio towards a more oxidized state. These changes were confined to the advanced stage of plant–pathogen interaction, suggesting the failure of regulatory mechanisms balancing NAD(H) and NADP(H) pool sizes and redox states in the early stage of infection. In the infected leaves, the NADPH pool was affected more dramatically and much earlier than in NAD+, supporting the predominant role of NADP(H) in defining the local response to infection.

Besides its defense-related redox functions, NADPH is involved in signaling reactions through reactive oxygen species (ROS), nitric oxide, and plant hormones [28], whose activation shortly after inoculation can decrease the NADPH content. NAD(H) and NADP(H) showed the same changing patterns of redox ratio in both infected and systemic leaves, but the nucleotide content changes leading to this occurred later in the latter. This suggests that the altered [NADH]/[NAD^+^] and [NADPH]/[NADP^+^] ratios are not only a consequence of metabolic adjustment to local infection, but also a potential mechanism to mediate the systemic response.

The NADP(H)-linked enzymes, coupled to central carbon metabolism, can influence the availability of NADP(H). In this study, the gene expression of the NADP^+^-linked *ICDH* did not change in the *Psl*-infected plants. Although the ICDH activity increased at the advanced stage of *Psl*-cucumber interaction [43], in this experimental setup, the NADPH-generating ICDH activity was likely insufficient to counteract the reduction of the [NADPH]/[NADP^+^] ratio in the infected plants. Interestingly, in L5 + *Psl*, the depletion of NADPH, which drives the anabolic reactions, was delayed, only being observed at 5–7 day; whereas in L3 + *Psl*, it lasted from 1 to 7 day. Although the NADPH-generating function of ICDH overlaps with other enzymes, e.g., G6PDH has been shown to be induced in L3 + *Psl*, it cannot be entirely replaced under increased H_2_O_2_ production, which typically occurs in infected tissues [44].

The chloroplast 3PGAD could also influence the NADPH pool in the *Psl*-infected plants. The expression of chloroplast *3PGAD* followed different patterns in response to *Psl* in the infected and systemic leaves. In the L3 + *Psl* leaves, it was upregulated only at 7 day, along with full disease symptom development, and in the L5 + *Psl*—at 2 day. The unchanged expression of *3PGAD* in the early phase of pathogenesis (1–2 day), accompanied by downregulation of *RuBisCO_LU_*, allowed redistribution of the reducing power in the chloroplasts by decreasing the NAD(P)H consumption by 3PGAD; therefore, favoring activation of defense responses in L3 + *Psl*. In the systemic leaves, however, its increased transcript accumulation at 2 day, concomitant with *RuBisCO_LU_*, was associated with the maintenance of high photosynthetic activity [1]. Interestingly, the chloroplast 3PGAD, apart from participating in photosynthetic CO_2_ fixation, is involved in plant innate immune response. In *Arabidopsis* infected with *P*. *syringae* pv *tomato*, 3PGAD isoforms act as negative regulators of plant immune responses, and the transcription of photosynthetic *3PGAD* was downregulated during PTI triggered by the pathogen [45].

### 3.2. Carbohydrate Composition and Metabolism

Pathogen infection often leads to changes in the distribution and metabolism of carbohydrates in plants. It is assumed that in the infected source leaves, the biotrophic pathogens’ demand for nutrients, mainly carbohydrates, creates an additional sink, and cell-wall acid invertase is a crucial factor in determining the sink strength in plants with sucrose as the primary translocated sugar [46,47,48]. According to Roitsch et al. [49], the induction of cell wall invertase can be interpreted as a source-to-sink conversion of the pathogen-colonized areas. The pathogen-induced decrease in *AcINV* expression and increase in *AlINV* expression described here did not confirm the results obtained for other pathosystems, which showed the induction of AcINV or a consistent stimulation of both AcINV and AlINV after infection [50,51]. Although the expression of *AcINV* was downregulated in the infected cucumber leaves, potentially limiting the availability of hexoses in the apoplastic space, the *Psl* needs for carbohydrates are likely to be met by other mechanisms. In *Arabidopsis* infected by *P*. *syringae* pv *tomato*, sugar efflux transporter SWEETs genes were induced, indicating that the pathogen used a different strategy to access carbohydrates from the plant [52]. In our study, the decreased expression of *AcINV* was accompanied by a progressive decline in the glucose/sucrose ratio in the infected leaves, and a low hexose/sucrose ratio was suggested to trigger the senescence associated with PR gene expression [53]. The lowered glucose/sucrose ratio, together with decreased expression of *RuBisCO_LU_* and the previously reported decline of photochemical activity of PSII and photosynthetic gas exchange at the advanced stage of infection development [1], further support the hypothesis that hemibiotrophic pathogens induce senescence in the infected organs, as they switch from biotrophy to necrotrophy [54].

*AlINV*, whose transcripts accumulated in the *Psl*-infected cucumber leaves, is involved in plant growth, development, carbon distribution, and carbohydrate signaling [55,56]. In *Arabidopsis*, alkaline/neutral invertases contribute to respiration providing glucose to hexokinase, which produces ADP, potentially used for ATP regeneration, supporting energy-demanding defense processes [57]. Our results correspond to this scenario; as in the L3 + *Psl* leaves, increased expression of *AlINV* was concomitant with declined sucrose and glucose contents, enhanced *HK* expression, and a rise in the respiration rate, which increases the consumption of photoassimilates. These adaptive regulations were not observed in the L5 + *Psl* leaves, which seemed to maintain photosynthesis [1] and respiration homeostasis.

Metabolic release of hexoses by invertases may play a regulatory role through carbohydrate signaling, in addition to the carbon and energy supply for both the plant and the pathogen. It was shown that the increased relative proportion of fructose amongst the total pool of sucrose, glucose, and fructose plays a decisive role in tomato defense against *Botrytis cinerea* [58]. We found that the percentage of fructose in the total pool of these soluble carbohydrates increased in L3 + *Psl* at 7 day in the absence of glucose and sucrose accumulation, implying differential roles of glucose and fructose in local defense against *Psl* or different use of these hexoses by the pathogen. No changes in the relative fructose content in the total sucrose, glucose, and fructose pool were found in L5 + *Psl*, indicating differential adjustments of these sugars in the pathogen-free systemic leaves.

The glucose content decrease in L3 + *Psl* leaves was paralleled by an increased glucose-6-phosphate concentration, G6PDH activity, and *HK* and *G6PDH* expression. The phosphorylation of glucose, catalyzed by HK, is pivotal for all metabolic pathways. In addition to this metabolic function, HK acts as a glucose sensor and represses the expression of photosynthetic genes in the nucleus of the source tissues. It also mediates the interactions of sugars with hormones and, together with abscisic acid (ABA), regulates photosynthetic activity through stomata closure [53,59]. The accumulation of *HK* transcripts and glucose-6-phosphate in L3 + *Psl*, and our previous reports on *Psl*-induced ABA content increase [60] accompanied by net photosynthesis and transpiration decrease [1], indicate that in these *Psl*-infected cucumber leaves, HK may have been involved in regulating the photosynthetic activity through this mechanism.

The turnover of glucose-6-phosphate, a substrate of the OPPP and intermediate of glycolysis, depends on the activity of HK and G6PDH. The latter is not only a rate-limiting enzyme in the OPPP, but also plays a role in response to pathogen infection. For example, in tobacco infected by *Phytophthora nicotianae*, G6PDH activity increased in the resistant cultivar but not in the susceptible one [61]. A primary role in the regulation of G6PDH is attributed to the [NADPH]/[NADP^+^] ratio; when this ratio is high, the G6PDH activity decreases [62]. Consistent with the decreased [NADPH]/[NADP^+^] ratio in the L3 + *Psl* leaves, G6PDH was induced at transcript and enzyme activity levels favoring the production of NADPH by increased OPPP and therefore supporting the defense response. In the L5 + *Psl*, a similar relationship between the [NADPH]/[NADP^+^] ratio and G6PDH activity was found only at 7 day, suggesting that in the systemic leaves in which the [NADPH]/[NADP^+^] ratio is low during active biosynthetic processes, G6PDH may be associated with the mechanism of balancing growth and defense through regulation of OPPP.

The metabolism of glucose-6-phosphate may also yield trehalose-6-phosphate and trehalose, as the binding of glucose-6-phosphate to UDP-glucose produces trehalose-6-phosphate, further cleaved into trehalose. These metabolites are sucrose level indicators that help plants maintain their sucrose/starch ratio. They contribute to water and osmotic stress tolerance by regulating carbohydrates levels, ABA signaling, and stomatal conductance during stress [63]. Trehalose is also involved in plant–pathogen interactions, but the exact mechanism remains unclear. It is required for the infectivity of pathogens [64], and at the same time, exogenous trehalose elicits plant defense responses and endogenous trehalose signals microbial pathogen attack [65,66]. In our study, *Psl* induced trehalose accumulation, which was virtually absent from control leaves at 0 and 2 day. This response occurred in both infected and non-infected systemic leaves. As trehalose is essential for the survival and fitness of *P*. *syringae* in the phyllosphere [67], it cannot be excluded that most of the trehalose in the L3 + *Psl* leaves was derived from the bacteria and could be necessary for manipulating host carbohydrate metabolism in favor of the pathogen [64]. However, the changes in trehalose content in L3 + *Psl* were almost precisely mirrored by those in the L5 + *Psl* leaves, indicating that the plant may sense trehalose as a sign of pathogen attack, triggering metabolic responses in the infected and distal leaves.

There were significant differences in the availability of carbohydrates providing metabolic energy in L3 + *Psl* and L5 + *Psl*, and glucose and sucrose were the major discriminative carbohydrates in the leaves of the infected plants. While cucumber is a raffinose family oligosaccharides-transporting species, in the infected plants, carbon partitioning was shifted towards transportable carbohydrates, i.e., raffinose synthesized from sucrose. We speculate that raffinose accumulated in the L3 + *Psl* leaves, acting as a scavenger of ROS [68], could have protected plant cells from infection-induced oxidative damage and promoted defense responses [69]. Similarly, raffinose played a positive role in the protection of cucumber roots from root-knot nematode at the initial stage of infection [70]. Raffinose accumulated in L5 + *Psl*, apart from taking part in ROS homeostasis and protecting the cellular metabolism, has a transport role associated with supplying nutrients to the systemic leaves. The diminished sucrose concentration in L3 + *Psl* at the advanced stage of pathogenesis could also indicate that carbohydrates from the infected leaves were translocated to the younger leaves as the sucrose content increased in L5 + *Psl* at 5 day. This suggested that in L5 + *Psl*, raffinose could be partly degraded into sucrose and hexose; likely to provide energy and carbon for the systemic leaves during the impaired metabolic activity of the infected ones.

The accumulation of raffinose in L3 + *Psl*, but not in L5 + *Psl*, was concomitant with the increased *myo*-inositol and glucose-6-phosphate contents and HK expression. A positive correlation between the contents of raffinose family oligosaccharides and *myo*-inositol, which plays a cofactor-like role in the biosynthesis of these sugars, was found in *Pisum sativum* seeds and transgenic *Solanum tuberosum* leaves [71,72]. *Myo*-inositol, synthesized from glucose-6-phosphate, is a precursor of many plant metabolites, including inositol-containing signaling molecules, such as phosphatidylinositols, implicated in plant–pathogen interactions [73]. A connection between inositol phosphate metabolism and basal resistance to viral, fungal, and bacterial pathogens, including *P*. *syringae*, was described in *Arabidopsis* [74]. Moreover, *myo*-inositol can regulate ROS-induced programmed cell death (PCD), a hallmark of HR-based immunity in plants [75]. We suggest that the relationship between *myo*-inositol and carbohydrate metabolism in L3 + *Psl* supports the hypothesis that *myo*-inositol, sucrose, and raffinose operate as metabolic mediators or signals in a regulatory circuit involved in the coordination of cellular processes in infected plants [73].

In our previous study, we showed that, due to the re-balancing of ascorbate and glutathione pools, chloroplasts in the systemic leaves of the *Psl*-infected plants generated a specific redox signature. This was suggested to be a regulatory element in integrating photosynthesis and the redox regulation of stress [1]. Metabolic processes such as photosynthetic carbon fixation, starch metabolism, fatty acids, and amino acids biosynthesis are known to be regulated by redox mechanisms, and signaling pathways reporting the cytosolic sugar status to the chloroplasts have also been identified [76]. Therefore, although the systemic metabolic responses were partly similar to those in the infected leaves, the specific chloroplast-related redox signature [1] and sugar status could be involved in a mechanism allowing the maintenance of photosynthesis in the L5 + *Psl* leaves. Except for the metabolic regulations, in the L3 + *Psl*, the pathogen negatively impacts photosynthesis by reducing the photosynthetically active leaf area as the disease progresses and the necrotic symptoms develop (7 day).

### 3.3. Malate- and Oxaloacetate-Related Metabolic Changes

The accumulation of oxaloacetic and malic acids, as well as increased activities and gene expression of FUM and l-MDH, accompanied by an increase in respiration rate, indicated that *Psl* induced the TCA cycle in the host. Malate is the key intermediate of plant respiration [77]; therefore, the higher malic acid concentration in L3 + *Psl* may support respiratory processes. The malic acid concentration increase in L3 + *Psl* was likely a consequence of PEPC upregulation, as it was paralleled by an increased expression of PEPC. In C3 plants, PEPC has often been described as replenishing the TCA cycle intermediates, but it is also a mediator of malate homeostasis under stress, supplies carbon skeletons for ammonium assimilation, and is involved in stomatal conductance and fixation of respired CO_2_ [78]. The *Psl*-induced stress conditions in the L3 + *Psl* leaves were manifested by decreased intracellular CO_2_ concentration, stomatal conductance, and net photosynthesis rate [1], as well as downregulation of *RuBisCO_LU_* expression. The lack of carboxylase activity of RuBisCO in these leaves could be compensated by PEPC recapturing the respired CO_2_, followed by the release of CO_2_ by the malic enzyme, which provides CO_2_ for the Calvin cycle [79]. The anapleurotic flux via PEPC, replenishing the TCA cycle intermediates to maintain the functionality of the cycle, could also be advantageous for the infected plants, which may have increased the metabolic building blocks and energy demand for defense responses [80]. The malic acid pool in the L3 + *Psl* leaves was also regulated by FUM converting fumarate to malate and NAD^+^-dependent l-MDH catalyzing the interconversion of malate to oxaloacetate, whose activities and gene expression levels were increased. While in L3 + *Psl*, the malic acid content, l-MDH, and FUM were positively related throughout the experiment, in L5 + *Psl*, this was observed only in the late phase of infection.

Apart from its critical role in the cellular energy metabolism, the TCA cycle is embedded in a network of metabolic pathways potentially involved in plant response to pathogens. These other functions require non-cyclic flux modes of carboxylic acid metabolism and the input of TCA intermediates, to compensate for the carbon loss from the cycle [81]. One important non-cyclic flux mode is involved in regulating the redox balance and requires a flux from oxaloacetate to malate in the mitochondrion. In spinach, up to 25% of the NADH generated from malate oxidation was exported via the malate-oxaloacetate shuttle to extra-mitochondrial processes requiring reducing power [82]. This mechanism could explain the increase in NADH content observed in the *Psl*-infected cucumber plants. The comparison of the changes in oxaloacetic and malic acid concentrations and l-MDH and FUM activities and gene expression levels suggests that in L3 + *Psl* and L5 + *Psl*, biotic stress induced different flux modes within the TCA cycle, depending specifically on the metabolic demand of the cells. Malate-oxaloacetate shuttles, acting in combination with different l-MDH isoforms located in plastids, peroxisomes, and the cytosol, are robust systems enabling the indirect transport of reducing equivalents between various cellular compartments [83]. Therefore, the malate content in the cucumber leaves may have also been influenced by non-mitochondrial l-MDH enzymes.

The accumulation of malic acid in L3 + *Psl* may also have contributed to the sustained availability of nutrients throughout the *Psl* infection. Malate is the preferred carbon source for *P*. *syringae* [84], and dicarboxylates were essential for the virulence of *P*. *syringae* pv *tomato* on *Arabidopsis* [85]. Thus, the metabolic reprogramming that is a consequence of the activated plant defense may simultaneously serve pathogen nutrition.

## 4. Materials and Methods

### 4.1. Plant Material, Bacterial Infection, and Sample Collection

Cucumber plants (*Cucumis sativus* L.) cv Cezar were grown in a growth chamber, in soil, at 23 °C, under irradiance of 350 μmol m^–2^ s^–1^ and a 16/8-h (day/night) photoperiod. Four-week-old plants were inoculated with *Psl* (isolate No. IOR 1990, Bank of Plant Pathogens, Poznań, Poland). The fully-expanded third true leaves from the base of the plants were inoculated with bacterial suspension, adjusted to 10^7^ cfu cm^–3^ (L3 + *Psl*) or treated with sterile distilled water (control, L3) using a needleless syringe (Figure A1). The fifth leaves were non-inoculated and collected from control (L5) or inoculated plants (L5 + *Psl*). The leaves were collected 3–4 h after the beginning of the light period (three plants were used to prepare one leaf sample) and analyzed on 0, 1, 2, 5, and 7 days after inoculation (day), as described earlier [1].

### 4.2. Measurements

#### 4.2.1. Respiration Rate Analysis

Respiration rate was measured as a decrease in O_2_ concentration using a Clark electrode (Hansatech Instruments Ltd., Norfolk, UK) in a LD/2 chamber connected to a data reader device CB1D (Hansatech). Measurements were made on discs (diameter of 5 cm), cut from cucumber leaves, under 21% O_2_, 350 ppm CO_2_ in a closed system, at a temperature of 25 °C.

#### 4.2.2. Determination of Nicotinamide Adenine Dinucleotides

Leaf samples (0.25 g) were homogenized in 1 cm^3^ of 0.1 M HCl for NAD^+^ or NADP^+^ determination, or in 0.1 M NaOH for NADH or NADPH determination. The homogenates were heated at 100 °C for 5 min, cooled, and centrifuged (10,000× *g*, 10 min, 4 °C). The supernatants were neutralized with 0.1 M NaOH and 0.1 M HCl, respectively, and centrifuged. The quantitative determination of nicotinamide adenine dinucleotides was based on the protocol of Gibon and Larher [86]. This spectrophotometric enzyme cycling assay involves 3-(4,5-dimethylthiazolyl-2)-2,5-diphenyltetrazolium bromide (MTT) as the final electron acceptor, and phenazine ethosulfate (PES) as an electron carrier in a reaction based, either on the conversion of ethanol to acetaldehyde, catalyzed by alcohol dehydrogenase (EC 1.1.1.1) for NAD(H) determination, or the conversion of glucose-6-phosphate to 6-phosphogluconolactone by glucose-6-phosphate dehydrogenase (G6PDH, EC 1.1.1.49) for NADP(H) determination. The reduction rate of MTT is proportional to the concentration of nicotinamide adenine dinucleotide. The contents of nicotinamide adenine dinucleotides were expressed in nmol g^–1^ FW.

#### 4.2.3. Enzyme Assays

For fumarase (FUM, EC 4.2.1.2) and G6PDH activity determination, leaf tissue was ground in 50 mM Tris-HCl, pH 7.7; and for malate dehydrogenase (l-MDH, EC 1.1.1.37) activity assay—in 50 mM K-phosphate buffer, pH 7.5. Fumarase activity was determined according to Bergmeyer et al. [87] by monitoring the formation of fumaric acid via the catalytic dehydration of malic acid at 240 nm and expressed in nmol malic acid min^−1^ mg^−1^ protein (ε = 2.25 mmol^–1^ dm^3^ cm^–1^). The activity of l-MDH, which converts oxaloacetate to l-malate, was assayed as the rate of oxaloacetate-dependent oxidation of NADH [88]. The change in absorbance at 340 nm is proportional to the enzymatic activity of l-MDH. Enzyme activity was expressed in nmol NADH min^–1^ mg^–1^ protein (ε = 6.22 mmol^–1^ dm^3^ cm^–1^).

G6PDH catalyzes the conversion of glucose-6-phosphate to 6-phosphogluconolactone and transfers one electron to NADP^+^ producing one NADPH. To determine the G6PDH activity, the increase in NADPH concentration was measured as the absorbance increase at 340 nm [89] and expressed in nmol NADPH min^−1^ mg^−1^ protein (ε = 6.22 mmol^–1^ dm^3^ cm^–1^).

#### 4.2.4. Determination of l-Malic and Oxaloacetic Acids

For the quantitative determination of l-malic acid, leaves were homogenized in triple distilled water. An enzymatic test kit based on the l-MDH-catalyzed oxidation of l-malic acid to oxaloacetate by NAD^+^ was used (Boehringer Mannheim/R-Biopharm, Darmstadt, Germany), and l-malic acid was determined according to the manufacturer’s recommendations. The amount of NADH formed is stoichiometric to the amount of l-malic acid.

For the determination of oxaloacetate, leaf extracts were prepared in 4% HClO_4_ [90]. The content of oxaloacetate was measured according to Wahlefeld [91]. The assay was based on the conversion of oxaloacetate to malate by NADH in the presence of l-MDH, and the amount of NADH oxidized to NAD^+^ is a mole for mole equivalent to substrate conversion. The contents of l-malic and oxaloacetic acids were expressed in mg g^–1^ FW.

#### 4.2.5. Quantification of Sucrose, d-Fructose, d-Glucose, and Glucose-6-Phosphate

Leaf samples (100 mg) were ground in liquid nitrogen, homogenized in 5 cm^3^ of warm 80% ethanol, and centrifuged (15,000× *g*, 10 min, 4 °C). The extraction procedure was repeated twice. The extracts were pooled, dried under vacuum at 37 °C, suspended in 1 cm^3^ of triple distilled water, and used for sugar assays. An enzymatic test kit was used according to the manufacturer’s protocol, to determine d-glucose, d-fructose, and sucrose contents (Boehringer Mannheim/R-Biopharm, Darmstadt, Germany). Invertase (EC 3.2.1.26) catalyzes the irreversible hydrolysis of sucrose to free glucose and fructose. In this assay, the d-glucose concentration is determined before and after the enzymatic hydrolysis of sucrose by invertase, and d-fructose is assayed subsequently to the determination of d-glucose. Carbohydrate contents were expressed in mg g^–1^ FW.

The glucose-6-phosphate concentration in cucumber leaves was assayed spectrophotometrically, based on the rate of G6PDH-catalysed reduction of NADP^+^ to NADPH by glucose-6-phosphate, converted to 6-phosphogluconolactone, according to [90]. Leaf extracts were prepared in 4% HClO_4_. The concentration of glucose-6-phosphate was expressed in μg g^–1^ FW.

#### 4.2.6. Determination of Trehalose, Raffinose, and Myo-inositol by GC-MS

Leaf samples (1 g), ground with liquid nitrogen, were sequentially extracted with hexane, diethyl ether, and methanol. For the determination of trehalose, raffinose, and *myo*-inositol, the residue obtained after extraction in diethyl ether was further extracted with methanol and dried under vacuum at 90 °C. The pellets were resuspended in 0.5 cm^3^ pyridine, and the metabolites were derivatized by adding 0.05 cm^3^ *N*,*O*-bis(trimethylsilyl)trifluoroacetamide containing 1% trimethylchlorosilane. Extracts were analyzed with an HP 6890 Gas Chromatograph equipped with a mass selective detector MSD 5973 (Agilent Technologies, Santa Clara, CA, USA), with an electronic autosampler 7693A ALS system, electronic pressure control, and split/splitless injector (Agilent Technologies, Santa Clara, CA, USA). The injector worked in a split 1:50 mode at 250 °C. The volume of the sample introduced into the injector was 1 μL. The transfer line temperature was 280 °C. Separation was performed on HP-5ms (30 m × 0.25 mm; 0.25 μm film thickness) fused silica column, with a helium flow rate of 1 cm^3^ min^–1^ [1].

#### 4.2.7. Gene Expression Analysis

RT-qPCR was used to detect changes in the gene expression of ribulose-l,5-bis-phosphate carboxylase/oxygenase (*RuBisCO*, EC 4.1.1.39) large subunit (*RuBisCO_LU_*), phosphoenolpyruvate carboxylase (*PEPC*, EC 4.1.1.31), l-MDH, glyceraldehyde-3-phosphate dehydrogenase (*3PGAD*, EC 1.2.1.12), *FUM*, isocitrate dehydrogenase (*ICDH*, EC 1.1.1.42), *G6PDH*, hexokinase (*HK*, EC 2.7.1.1), acid and alkaline invertase (*AcINV*, *AlINV*, EC 3.2.1.26), and chloroplastic thioredoxin M3 (*TrxM3*). Total RNA extraction from cucumber leaves, RT-qPCR assays, and relative quantification of RT-qPCR were according to [1]. Primer sequences for target and reference genes are shown in Table A2.

#### 4.2.8. Protein Determination

Protein concentration was determined according to Bradford [92], and bovine serum albumin was used as a standard.

#### 4.2.9. Statistical Analysis

The data were analyzed using the Kruskal-Wallis test (Statistica^®^ software, ver. 12, StatSoft, Inc., Tulsa, OK, USA). Data are given as mean values of 4–5 replicates obtained in 4–5 independent experiments, and a single sample was analyzed per treatment in each experiment. Sample variability is given as a standard deviation of the arithmetic average. Differences at *p* ≤ 0.05 were considered significant.

## 5. Conclusions

The *Psl*-induced metabolic signature differed in the infected and uninfected systemic cucumber leaves. *Psl* altered the [NADH]/[NAD^+^] and [NADPH]/[NADP^+^] ratios and the soluble carbohydrate composition of the leaf cells. Infection shifted the pyridine nucleotide redox ratios such that the [NADH]/[NAD^+^] ratio increased and [NADPH]/[NADP^+^] declined at the advanced stage of pathogenesis; however, the related [NADPH] and [NAD^+^] changes occurred earlier in the infected than systemic leaves. Sucrose and glucose contents and turnover, evolving differently in the *Psl*-infected leaves and the pathogen-free systemic ones, constituted the major discriminative elements of the *Psl*-induced metabolic signature in the infected plants. The soluble carbohydrate-related changes at the transcript and metabolic levels in the infected leaves could directly support local defense processes requiring metabolic energy and provide metabolic mediators or signals involved in the coordination of defense responses, as suggested for the regulatory circuit composed of *myo*-inositol, sucrose, and raffinose. They could also support bacterial nutrition throughout the infection cycle. In the systemic leaves, the responses that involve soluble carbohydrates were less pronounced and likely related to the mechanism of balancing growth and defense at the organism level.

The disease progress in the *Psl*-infected leaves was positively correlated with the respiration rate, supporting the hypothesis of a link between local photosynthesis downregulation in the infected tissues [1] and the activation of processes fueling defense processes [37]. Significant infection-responsive changes in the primary carbon metabolism indicated the induction of the TCA cycle and were related to malic acid content and metabolism. The accumulation of malic acid and mobilization of l-MDH and FUM activities in the systemic leaves mirrored those at the infection sites, but occurred later.

In general, *Psl* elicited metabolic changes common to the infected and non-infected leaves, the dynamics of which differed quantitatively and metabolic modifications specifically related to the local or systemic response. The metabolic acclimation of systemic leaves to local pathogen infection is involved in balancing the trade-off between growth and defense at the whole-plant level. It may also prime defense responses in the systemic leaves if they become infected by a pathogen.

## Figures and Tables

**Figure 1 ijms-23-12418-f001:**
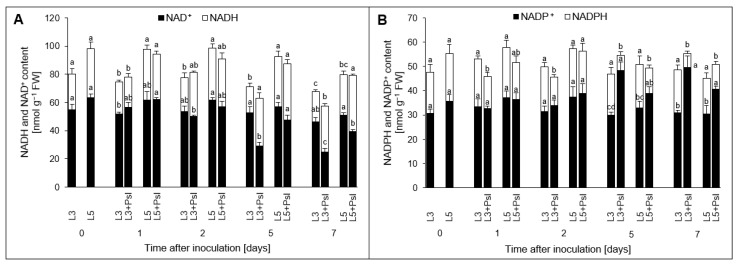
Nicotinamide adenine dinucleotide contents in cucumber leaves after *P. syringae* pv *lachrymans* infection: (**A**) NADH, NAD^+^ and (**B**) NADPH, NADP^+^. Values are means of four replicates (±SD). Different letters (a–d) indicate significant (*p* ≤ 0.05) differences between experimental variants within a given time point.

**Figure 2 ijms-23-12418-f002:**
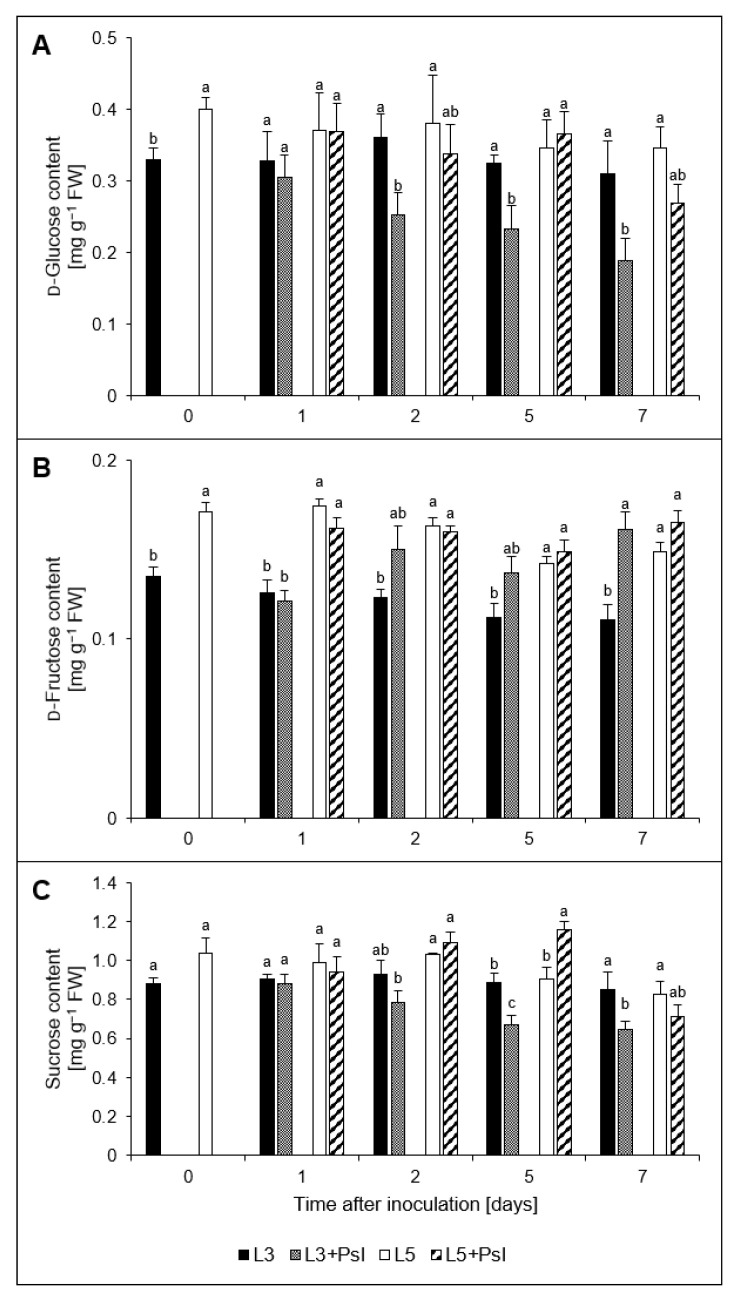
Changes in (**A**) d-glucose, (**B**) d-fructose, and (**C**) sucrose contents in cucumber leaves after *P. syringae* pv *lachrymans* infection. Values are means of five replicates (±SD). Different letters (a–c) indicate significant (*p* ≤ 0.05) differences between experimental variants at a given time point.

**Figure 3 ijms-23-12418-f003:**
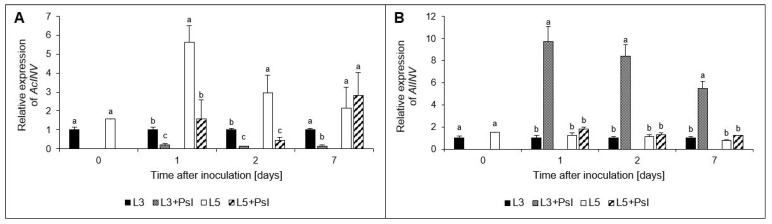
Relative expression of (**A**) acid invertase (*AcINV*) and (**B**) alkaline invertase (*AlINV*) genes in cucumber leaves after *P. syringae* pv *lachrymans* infection. Values are means of four replicates (±SD). Different letters (a–c) indicate significant (*p* ≤ 0.05) differences between experimental variants at a given time point.

**Figure 4 ijms-23-12418-f004:**
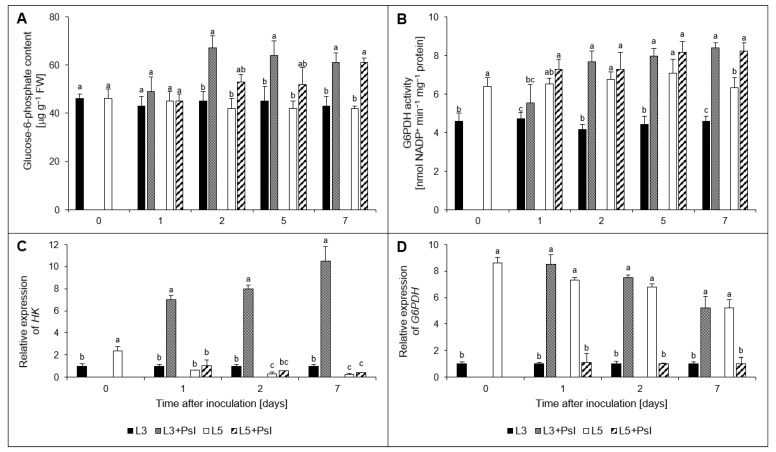
Infection-induced changes in (**A**) glucose-6-phosphate content, (**B**) glucose-6-phosphate dehydrogenase (G6PDH) activity, and the relative expression of (**C**) hexokinase (*HK*) and (**D**) glucose-6-phosphate dehydrogenase (*G6PDH*) genes in cucumber leaves. Values are means of four replicates (±SD). Different letters (a–c) indicate significant (*p* ≤ 0.05) differences between experimental variants at a given time point.

**Figure 5 ijms-23-12418-f005:**
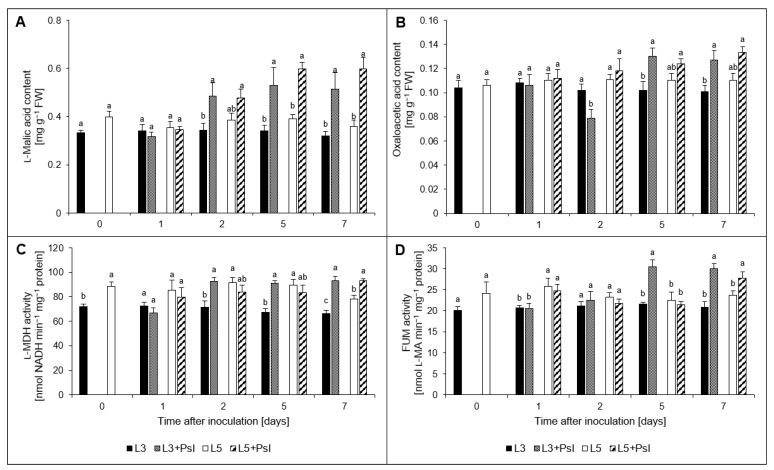
Changes in (**A**) l-malic acid content, (**B**) oxaloacetic acid content, (**C**) l-malic acid dehydrogenase (l-MDH) activity, and (**D**) fumarase (FUM) activity in cucumber leaves after *P. syringae* pv *lachrymans* infection. Values are means of four replicates (±SD). Different letters (a–c) indicate significant (*p* ≤ 0.05) differences between experimental variants at a given time point.

**Figure 6 ijms-23-12418-f006:**
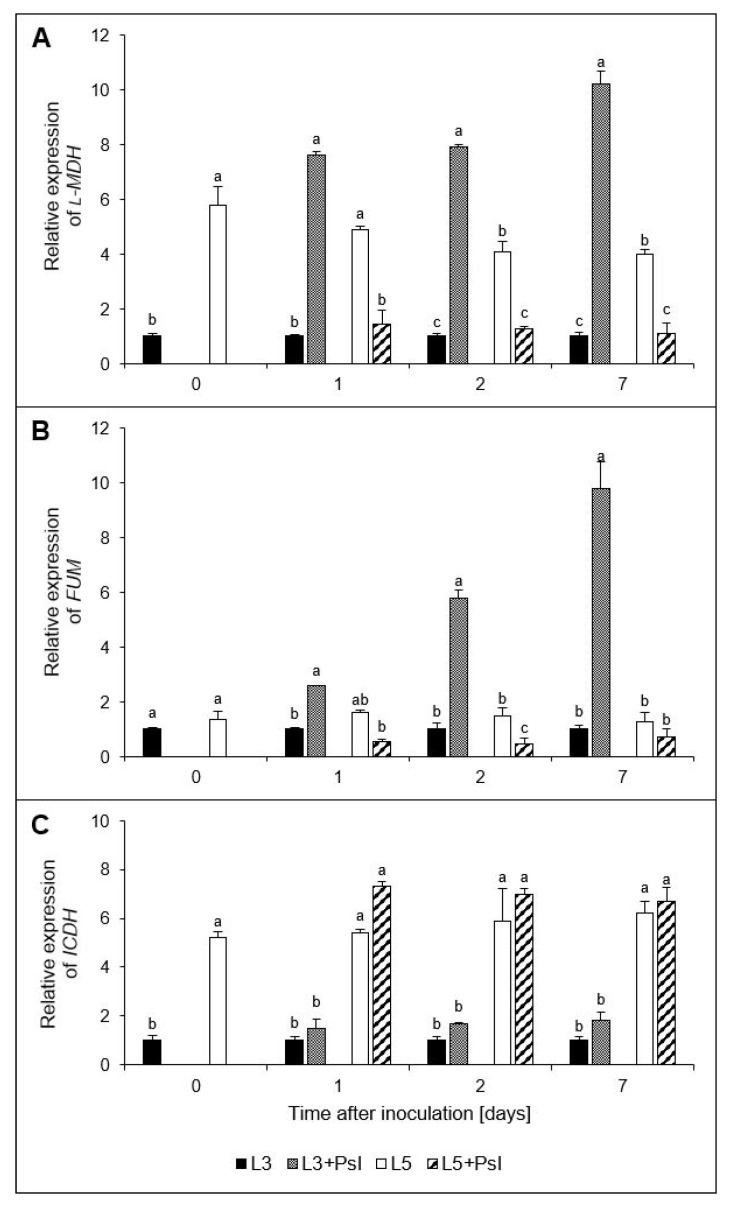
Relative expressions of (**A**) l-malic acid dehydrogenase (*l-MDH*), (**B**) fumarase (*FUM*), and (**C**) isocitrate dehydrogenase (*ICDH*) genes in cucumber leaves after *Psl* infection. Values are means of four replicates (±SD). Different letters (a–c) indicate significant (*p* ≤ 0.05) differences between experimental variants at a given time point.

**Figure 7 ijms-23-12418-f007:**
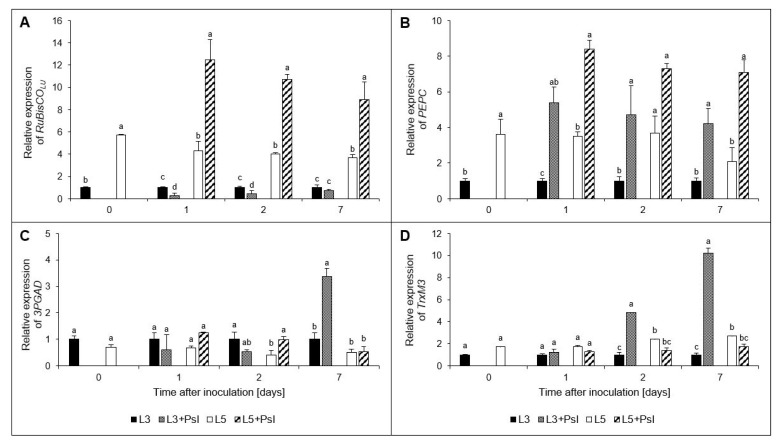
Relative expression of genes in cucumber leaves after *Psl* infection: (**A**) ribulose-1,5-bisphosphate carboxylase/oxygenase large unit (*RuBisCO_LU_*), (**B**) phosphoenolpyruvate carboxylase (*PEPC*), (**C**) chloroplast glyceraldehyde 3-phosphate dehydrogenase (*3PGAD*), and (**D**) chloroplast thioredoxin M3 (*TrxM3*). Values are means of four replicates (±SD). Different letters (a–d) indicate significant (*p* ≤ 0.05) differences between experimental variants at a given time point.

**Table 1 ijms-23-12418-t001:** Respiration intensity (μmol O_2_ m^–^^2^ s^–^^1^) in cucumber leaves after *Pseudomonas syringae* pv *lachrymans* infection.

Time (day)	L3	L3 + *Psl*	L5	L5 + *Psl*
0	1.976 ± 0.269 (b)		2.374 ± 0.212 (b)	
2	2.300 ± 0.354 (b)	3.589 ± 0.219 (a)	2.450 ± 0.272 (b)	2.344 ± 0.265 (b)
7	1.956 ± 0.235 (b)	2.926 ± 0.266 (a)	2.494 ± 0.306 (ab)	2.400 ± 0.267 (ab)

Values are means of four replicates (±SD). Different letters (a–b) indicate significant (*p* ≤ 0.05) differences between experimental variants within a given time point. “day”—days after inoculation.

**Table 2 ijms-23-12418-t002:** Infection-induced changes in relative metabolite contents in cucumber leaves after *Pseudomonas syringae* pv *lachrymans* infection.

Metabolite	Time (day)	L3	L3 + *Psl*	L5	L5 + *Psl*
Raffinose	0	1.0 ± 0.1 (a)		0.2 ± 0.1 (b)	
	2	1.0 ± 0.2 (c)	1.9 ± 0.1 (b)	0.1 ± 0.02 (d)	2.8 ± 0.1 (a)
	7	1.0 ± 0.2 (c)	4.0 ± 0.3 (a)	0.5 ± 0.1 (d)	3.0 ± 0.1 (b)
Trehalose	0	-		-	
	2	-	0.6 ± 0.2 (a)	-	0.3 ± 0.1 (b)
	7	0.4 ± 0.1 (b)	0.8 ± 0.2 (a)	0.4 ± 0.1 (b)	0.7 ± 0.1 (a)
*Myo*-inositol	0	3.5 ± 0.1 (b)		6.9 ± 0.2 (a)	
	2	3.7 ± 0.5 (b)	8.4 ± 0.6 (a)	6.5 ± 0.1 (ab)	6.0 ± 0.5 (ab)
	7	3.8 ± 0.5 (b)	7.3 ± 0.4 (a)	6.2 ± 0.7 (ab)	6.2 ± 1.0 (ab)

Values are means of four replicates (±SD). Different letters (a–d) indicate significant (*p* ≤ 0.05) differences between experimental variants within a given time point. “-“ not detected; day—days after inoculation.

## Data Availability

Not applicable.

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
