# Peer review of "Alterations in Primary Carbon Metabolism in Cucumber Infected with Pseudomonas syringae pv lachrymans: Local and Systemic Responses"

_ijms, 2022, doi:10.3390/ijms232012418_

Round 1

Reviewer 1 Report

1. Why the experimental samples didn’t have replicates?

2. Please describe more detail the procedures of the experiments, for example “how to collect the samples of leaves?”

Author Response

1. Why the experimental samples didn’t have replicates?

Response: The following explanation has been added to Section 4.2.9.: “Data are given as means of 4-5 replicates obtained in 4-5 independent experiments, and a single sample was analyzed per treatment in each experiment”.

2. Please describe more detail the procedures of the experiments, for example “how to collect the samples of leaves?”

Response: Leaf sample collection has been described in Section 4.1. We have also added a new figure (Appendix: Figure A1) showing the experimental design and the disease symptom progression.

3. Some of the experimental methods have been described more clearly (Sections 4.1, 4.2.1, 4.2.3, 4.2.5). 

4. The manuscript has been edited by a native English-speaking editor.

Reviewer 2 Report

This manuscript describes a few metabolic and transcriptional changes in cucumber after inoculation of Pseudomonas syringae pv lachrymans, at local and systemic levels.

The intensity of the symptoms and the pathogen proliferation should be quantified and shown in the manuscript, so that a link between symptom intensity and metabolic changes can be established.

For each enzyme analyzed, the activity of the enzyme (substrate and product) should be mentioned.

Author Response

1. The intensity of the symptoms and the pathogen proliferation should be quantified and shown in the manuscript, so that a link between symptom intensity and metabolic changes can be established.

Response: The disease severity progression given as the percentage of the total leaf area covered with symptoms, the changes in pathogen proliferation (Log CFU g-1 FW) as well as angular leaf spot disease symptoms were shown in Figure A1 added to the Appendix of the revised version of the manuscript.

2. For each enzyme analyzed, the activity of the enzyme (substrate and product) should be mentioned

Response:

The missing substrate and product were mentioned for glucose-6-phosphate dehydrogenase (Section 4.2.3) and invertase (Section 4.2.5).

3. The Introduction has been revised to justify the study.

4. Some of the experimental methods have been described more clearly (Sections 4.1, 4.2.1, 4.2.3, 4.2.5). 

5. The manuscript has been edited by a native English-speaking editor.

Changes to the manuscript have been highlighted in yellow in the copy of the revised version.

Reviewer 3 Report

This study examined several metabolic changes upon comparing the P. syringae pv lachrymans (Psl)-infected cucumber leaves and the uninoculated control. The result of the manuscript is straightforward to follow. However, the significance of the manuscript may need polish. Herein, I provided several suggestions for the authors.

1, The authors claimed that in their previous study, the photosynthesis declines upon Psl-infection were confined to the infected leaves, whereas the pathogen-free systemic leaves maintained their photosynthetic capacity [1]. However, in this manuscript, the author claims that "Metabolic changes in the systemic leaves paralleled the local responses but occurred later (e.g., malic acid content and metabolism, glucose-6-phosphate accumulation and G6PDH activity) or showed a specific induction pattern (e.g., hexose contents). Therefore, they may be part of the global effects of local infection on plant metabolism and also represent specific responses triggered by signals from the infected leaves priming their protective/acclimation status". These two conclusions seem to be a paradox. If the local response affects the systemic metabolism at the molecular level, how does the photosynthetic capacity in the systemic remains the same? The authors may need to provide a detailed explanation.

2, Many biochemical pathways, primarily carbon metabolic, are involved in plant respiration. Therefore, it is better to provide a blueprint of the Psl-induced changes and further address the pathways described in the manuscript. Without this information, it looks like the authors randomly chose to measure the pathways.

3, The conclusion "Therefore, they may be part of the global effects of local infection on plant metabolism and also represent specific responses triggered by signals from the infected leaves priming their protective/acclimation status" should be polished. Considering that the metabolic changes in the systemic leaves paralleled the local responses but occurred later, is it possible that the movement of this metabolism from the local to systemic leaves matters? More evidence is needed to claim the "signaling" matters.

Author Response

1. The authors claimed that in their previous study, the photosynthesis declines upon Psl-infection were confined to the infected leaves, whereas the pathogen-free systemic leaves maintained their photosynthetic capacity [1]. However, in this manuscript, the author claims that "Metabolic changes in the systemic leaves paralleled the local responses but occurred later (e.g., malic acid content and metabolism, glucose-6-phosphate accumulation and G6PDH activity) or showed a specific induction pattern (e.g., hexose contents). Therefore, they may be part of the global effects of local infection on plant metabolism and also represent specific responses triggered by signals from the infected leaves priming their protective/acclimation status". These two conclusions seem to be a paradox. If the local response affects the systemic metabolism at the molecular level, how does the photosynthetic capacity in the systemic remains the same? The authors may need to provide a detailed explanation.

A detailed explanation has been provided in the last paragraph of Section 3.2. Moreover, in the Introduction, in the paragraph referring to the results of our previous work, we explained that pathogen-induced decline in photochemical activity of PSII, stomatal conductance and photosynthetic gas exchange at the advanced stage of pathogenesis were related to redox regulations (Introduction, L 149-150).

2. Many biochemical pathways, primarily carbon metabolic, are involved in plant respiration. Therefore, it is better to provide a blueprint of the Psl-induced changes and further address the pathways described in the manuscript. Without this information, it looks like the authors randomly chose to measure the pathways.

We provided an explanation of why the parameters described in the manuscript were chosen to measure (Introduction, L 155-156). To precisely address the changes, we used the term ”respiration rate”, as this parameter was measured (Discussion, L396).

3. The conclusion "Therefore, they may be part of the global effects of local infection on plant metabolism and also represent specific responses triggered by signals from the infected leaves priming their protective/acclimation status" should be polished. Considering that the metabolic changes in the systemic leaves paralleled the local responses but occurred later, is it possible that the movement of this metabolism from the local to systemic leaves matters? More evidence is needed to claim the "signaling" matters.

This conclusion has been changed (Abstract, L 33-40). Accordingly, Section 5 has been modified (Section 5, L 669 and L676-679).

4. The Introduction and Discussion have been revised. Four references have been added.

5. Some of the experimental methods have been described more clearly (Sections 4.1, 4.2.1, 4.2.3, 4.2.5). One new figure showing the disease severity progression given as the percentage of the total leaf area covered with symptoms, the changes in pathogen proliferation (Log CFU g-1 FW), as well as angular leaf spot disease symptoms, has been added (Figure A1).

6. The manuscript has been edited by a native English-speaking editor.

Changes to the manuscript have been highlighted in yellow in the copy of the revised version.

Round 2

Reviewer 3 Report

My concerns were addressed adequately in the revised manuscript.

Though the manuscript has been edited by a native speaker, I strongly suggest authors go through the whole manuscript for potential grammar mistakes.  For instance, please provide the full names of the proteins/genes or pathways (TCA) in the abstract, even though they are widely used.